# Fluidized Bed Drying of Wheatgrass: Effect of Temperature on Drying Kinetics, Proximate Composition, Functional Properties, and Antioxidant Activity

**DOI:** 10.3390/foods12081576

**Published:** 2023-04-07

**Authors:** Ranjika Chakraborty, Piyush Kashyap, Ram Kaduji Gadhave, Navdeep Jindal, Shiv Kumar, Raquel P. F. Guiné, Rahul Mehra, Harish Kumar

**Affiliations:** 1Department of Food Engineering and Technology, Sant Longowal Institute of Engineering and Technology, Punjab 148106, India; 2Department of Food Technology and Nutrition, School of Agriculture, Lovely Professional University, Punjab 144001, India; 3Food Science & Technology MMICT&BM(HM), Maharishi Markandeshwar (Deemed to be University), Ambala 133207, India; 4CERNAS Research Centre, Polytechnic Institute of Viseu, Campus Politécnico, 3504-510 Viseu, Portugal; 5Amity Institute of Biotechnology, Amity University Rajasthan, Jaipur 303002, India

**Keywords:** wheatgrass, drying kinetics, antioxidant activity, functional properties, moisture ratio

## Abstract

Wheatgrass is a valuable source of nutrients and phytochemicals with therapeutic properties. However, its shorter life span makes it unavailable for use. So, storage-stable products must be developed through processing in order to enhance its availability. Drying is a very important part of the processing of wheatgrass. Thus, in this study, the effect of fluidized bed drying on the proximate, antioxidant, and functional properties of wheatgrass was investigated. The wheatgrass was dried in a fluidized bed drier at different temperatures (50, 55, 60, 65, 70 °C) using a constant air velocity of 1 m/s. With increasing temperature, the moisture content was reduced at a faster rate, and all drying processes took place during the falling rate period. Eight mathematical models under thin layer drying were fitted into the moisture data and were evaluated. The Page model was the most effective in explaining the drying kinetics of wheatgrass, followed by the Logarithmic model. The R^2^, chi-square, and root mean squared value for Page model was 0.995465–0.999292, 0.000136–0.0002, and 0.013215–0.015058, respectively. The range of effective moisture diffusivity was 1.23–2.81 × 10^−10^ m^2^/s, and the activation energy was 34.53 kJ/mol. There was no significant difference in the proximate composition of was seen at different temperatures. The total phenolic content (117.16 ± 0.41–128.53 ± 0.55 mgGAE/g), antioxidant activity (33.56 ± 0.08–37.48 ± 0.08% (DPPH), and FRAP (1.372 ± 0.001–1.617 ± 0.001 mgAAE/g) increased with the rise in temperature. A significant increase was observed in functional properties, except for the rehydration ratio, which decreased with rising temperature. The current study suggests that fluidized bed drying improves the nutritional retention of wheatgrass with good antioxidant activity and functional properties that can be used to make functional foods.

## 1. Introduction

Wheatgrass is a young plant freshly sprouted from bold wheat (*Triticumaestivum*) seeds and allowed to grow for a minimum period of 8–10 days [1]. It can be cultivated outdoors as well as indoors, but indoor cultivation is most common. It contains approximately 70% chlorophyll, and it has been reported that chlorophyll-rich young grasses may have anti-cancer components and can treat high blood pressure, obesity, cancers, diabetes, fatigue, gastritis, ulcers, skin problems, constipation, asthma, eczema, and anaemia [2]. Wheatgrass is rich in proteins; total dietary fibre; carbohydrates; vitamins such as A, C, E, niacin, riboflavin, and folic acid; minerals such as iron, calcium, magnesium, selenium, chlorophyll, and carotene; and antioxidants. It also contains 17 amino acids, among which eight of them are essential and a natural source of blood building [1,3]. Wheatgrass also has abundant antioxidants that can be beneficial in treating various diseases such as diabetes and cardiovascular diseases and proved to be a promising remedy for patients suffering from anaemia, cancer, eczema, constipation, kidney swelling, and the common cold [4].

Wheatgrass can be stored in the refrigerator for 2 to 3 weeks at 33 to 40 °F; however, some nutrients are lost during refrigeration [5]. Hence, it is better to consume the wheatgrass within a few days of harvesting [1]. Due to its high moisture content, it is a perishable item and will deteriorate within a short period of time. Therefore, some processing can help to preserve and commercialize the wheatgrass. Drying is the most favourable processing technique that can preserve the nutrients in the product. To prevent heat losses, drying should be carried out at the lowest possible temperatures. The high consumption rate of wheatgrass indicates its value and potential in the market, which requires high quality, easy availability, and a constant supply of raw materials. The drying process can decrease the moisture content, increase shelf life, and facilitate its use, thereby contributing to the supply, availability, and demand on the market [6]. Drying can reduce the water content up to a certain level at which spoilage due to microorganisms and degradation of chemical reactions can be controlled. There are various methods of drying which could possibly reduce the moisture content to the desired level. For example, hot air drying is one of the most common methods used for the drying of food products and involves simultaneous heat and mass transfer under a transitory regime. However, the main disadvantage of conventional hot air drying is that it requires a considerable amount of time, even at high temperatures, resulting in a serious loss of quality attributes, including flavour, colour, porosity, and bioactive compounds, as well as damage to the product itself [7]. Moreover, freeze drying could be a potential alternative to hot air drying that preserves the nutritional value and helps in maintaining colour and appearance. However, it is quite expensive and would not be cost-effective for larger scales of production [8]. Hence, the fluidized bed drying method is the best alternative method for valorising wheatgrass. This method has a number of advantages over the traditional method of convection drying in a monolayer configuration such as (1) the heat and the mass transfer rate is high as the particles and airflow have good contact, (2) the drying capacity is high because the ratio of the mass of air to the mass of the product is high, and (3) the bulk moisture content and temperature of particles is uniform as the mixing of particles in the bed is rigorous [9]. Sozzi et al. [7] used fluidized bed drying to dry blackberry waste, and they found the fibre types were more balanced and the pigments and colour of the blackberry were significantly retained. Fluidized bed drying of Bentong ginger also showed high antioxidant activity (89.2%) compared to oven drying and freeze drying and helped in preserving the 6-gingerol compound [8].

Mathematical modelling of the drying process and equipment is an integral part of drying technology [9,10]. The concept behind this modelling is based on its own set of mathematically derived equations which describes the drying operation [11]. Mathematical modelling is used to predict drying rate and efficiency under various conditions that can decide the best quality of the end product and reduce the processing time. Many mathematical models have come up with a detailed representation of the drying behaviour of agricultural commodities. These models are theoretical, semi-theoretical, and empirical models [12]. In imitating the drying process, semi-theoretical and empirical models become intuitive; in addition, their application becomes constrained only to the radius of drying conditions during the experiment. The root of the empirical method is experimental data and dimensionless analysis, in which the average moisture content and the drying time are interdependent of each other [6]. Semi-theoretical models are derived from the general solution to Fick’s second law through simplification. The Lewis (Newton) model, the Two-term model, the Page model, the Logarithmic model, the Modified Page model, the Henderson and Pabis model, and the Two-term exponential model are the most commonly used models for drying experiments of different agricultural produce [10,11,12]. 

Drying methods attempting to dehydrate and commercially preserve wheatgrass have resulted in poor product quality because of the product’s reduced nutritional quality. Pardeshiet al. [13] carried out drying methods such as forced air shade, solar drying with natural airflow, and shade and sun drying to predict an appropriate method of dehydration with the help of Page’s equation, which could ultimately help to retain some of the nutritional values of wheatgrass. However, so far as we know, no information has been published on the drying kinetics of wheatgrass using a fluidized bed drier to find the exact time and temperature that can effectively dehydrate wheatgrass without causing its valuable nutrients to deteriorate as well as help in improving its functional characteristics. Thus, the study’s objective is to investigate the effect of temperature and time on the drying kinetics of wheatgrass in a fluidized bed dryer that would yield a product comparable in quality, in terms of physicochemical and nutritional properties and antioxidant activity. The available moisture data were also fitted into various mathematical models available in the literature.

## 2. Materials and Methods

### 2.1. Wheatgrass Sample

Wheat seeds were procured from the local market of Longowal, Punjab, India. A wet cotton cloth was placed over the seeds and allowed to sprout for a period of 48 h after they were soaked in tap water for 24 h. The sprouted wheat seeds were then placed on perforated soil bed trays indoors. These sprouts were watered twice a day, and the trays were placed so that the young plants could receive normal airflow and sunlight. The sprouted seeds then began to develop into young leaf blades, which are called wheatgrass. The wheat grasses were harvested on the 10th day after the sprouting period. Harvested wheatgrass was chopped into small, equally sized pieces used for the drying experiment. Wheatgrass samples were placed in a hot air oven at 110 °C to determine initial moisture content.

### 2.2. Drying Procedure

A fluidized bed drier (Retsch-Model TG200, Haan, Germany) was used in this study to dry wheatgrass. A glass and cylindrical column was used in the drying chamber. At a fixed bed height and air velocity of 3 cm and 1 m/s, respectively, drying experiments were conducted at 5 °C intervals between 50 and 70 °C. The drying procedure was continued by subsequent weighing of the glass chamber and the wheatgrass sample at a regular interval of 10 min on a digital weighing balance. The drying process was carried out up to the point there was no change in readings of the weight of the sample with the glass chamber for three consecutive readings.

### 2.3. Mathematical Modeling

Table 1 summarizes eight simplified drying models that have been used to describe the drying kinetics of wheatgrass moisture ratios (MR); the models are derived from drying data, and their plots against drying times are presented. The moisture ratio and drying rate (DR) are determined by the following equations:(1)$$MR=\frac{M-Me}{Mo-Me}$$
where MR stands for the moisture ratio, M stands for the moisture content at time t on a dry basis, M_e_ is the equilibrium constant of moisture on a dry basis, and M_0_ commences moisture content on a dry basis.
(2)$$DR=\frac{M\left(t+dt\right)-Mt}{dt}$$
where M(t + dt) stands for the moisture content at time t + dt, Mt stands for the moisture content at time t, and dt is the time difference. Statistica 10 was used to calculate all non-linear regression data. To evaluate the fit quality of a specific model, the coefficient of determination (*R^2^*), reduced chi-square (*χ^2^*), and root mean square error (*RMSE*) was used. The *RMSE* and reduced chi-square can be calculated using the formulas:$$R^{2}=\frac{\sum_{i=1}^{N}{\left(MR_{ei}-MR_{e_{mean,\;i}}\right)}^{2}-{\left(MR_{pi}-MR_{p_{mean,\;i}}\right)}^{2}}{\sum_{i=1}^{N}{\left(MR_{ei}-MR_{e_{mean,\;i}}\right)}^{2}}$$
(3)$$\;\;\chi 2=\overset{N}{\underset{i=1}{\sum }}\frac{{\left(MR_{ei}-MR_{pi}\right)}^{2}}{N-Z}$$
(4)$$RMSE=\sqrt{\left[\frac{1}{N}\overset{N}{\underset{i=1}{\sum }}{\left(MR_{ei}-MR_{pi}\right)}^{2}\right]}$$
where *MR_ei_* is the experimental dimensionless moisture ratio and *MR_pi_* is the predicted dimensionless moisture ratio, *N* stands for the number of observations, and *Z* stands for the number of constants in the model. The higher value of *R^2^* and lower *RMSE* and *χ*^2^ values determine the model’s goodness of fit.

### 2.4. Effective Moisture Diffusivity Estimation

The effective moisture diffusivity was determined using Fick’s second law of diffusion at various air-drying temperatures. Assuming no shrinkage or temperature change has occurred, the general equation for the solution is as follows:(5)$$\frac{Mt-Me}{Mo-Me}=\frac{8}{\pi^{2}}\overset{\infty }{\underset{n=0}{\sum }}\frac{1}{\left(2n+1\right)}\exp{\left(-(2n+1\right)}^{2}\pi^{2\;}D_{eff}t\;÷4L^{2})$$
where:

*D_eff_* is the effective diffusivity in m^2^/s;

*L* is the half thickness of the slab in m;

*t* is the time in seconds.

Using logarithmic expressions, Equation (5) can be simplified into:(6)$$\ln\;MR=\ln\left(\frac{8}{\pi^{2}}\right)-\frac{\pi^{2}D_{eff}}{4L^{2}}\times t$$

The slope of the ln *MR* and *t* (drying time) plot is used to determine the effective moisture diffusivity using Equation (6):(7)$$Slope=-\frac{\pi^{2}D_{eff}}{4L^{2}}$$

### 2.5. Estimation of Activation Energies

An Arrhenius-type equation is used to calculate the activation energy, where temperature is a function of effective moisture diffusivity. It is given as
(8)$$D_{eff}=D_{eff}o\exp\;\left(-\frac{Ea}{RgTa}\right)\;$$
where:

*D_eff_o* is the effective moisture diffusivity for an infinite temperature in m^2^/s;

*Ea* is the activation energy for the diffusion of moisture in kJ/mol;

*Rg* is the universal gas constant (8.314 × 10^−3^ KJ mol^−1^ k^−1^);

*Ta* is the absolute temperature in k.

### 2.6. Characterization of Wheatgrass Powder

#### 2.6.1. Proximate Composition

The proximate composition of wheatgrass powder (carbohydrate, moisture, protein, fat, ash, and fibre) was determined according to the standard AOAC methods [14]. Kjeldahl was used for the determination of crude protein content using a 6.25 conversion factor. A gravimetric analysis of the lipid content was conducted following Soxhlet extraction. Acid/alkaline hydrolysis of insoluble residues estimated crude fibre content. We estimated the crude ash content through incineration at 550 °C in a muffle furnace. Using the difference method, the available carbohydrates were estimated.

#### 2.6.2. Chlorophyll Content

The determination of chlorophyll content was performed as per the method given by Pardeshi et al. [13]. One gram of wheatgrass powder sample was homogenized with ten millilitres of 80% acetone. The mixture was placed in a centrifuge at 5000 rpm for 5 min, and the filtrate was collected. The residue left after centrifugation was extracted again and again with 80% acetone until it became colourless. The residue was then diluted with 80% acetone, resulting in a final volume of 100 mL. The absorbance at 663 nm and 645 nm was determined using a spectrophotometer. The chlorophyll pigment content was calculated by using the equation:$$mg\;\mathrm{total}\;\mathrm{chlorophyll}\;\left(\mathrm{per}\;g\;\mathrm{tissue}\right)=20.2\;(A_{645})+8.02\;(A_{663})\times \frac{V}{1000}\times W$$
where A is denoted as the absorbance at a specific wavelength, V is denoted as the final volume of the chlorophyll extract in 80% acetone, and W is the weight of fresh tissue extract.

#### 2.6.3. Total Phenolic Content (TPC)

The TPC of dried wheatgrass was measured according to Folin–Ciocalteau’s reagent method [15]. A 100 μL sample of wheatgrass extract in DMSO (dimethyl sulfoxide) (1 mg/mL) was mixed in 250 μL of Folin—Ciocalteau reagent. Then, 1.5 mL of 20% Na_2_CO_3_solution was added after 5 min, and the solution was prepared up to 5 mL. After standing in the dark for two hours, the absorbance of this mixture was measured at 760 nm. Gallic acid was taken as standard. The total amount of phenols was measured in milligrams of gallic acid equivalent (mg GAE/g) of dried wheatgrass extract.

#### 2.6.4. DPPH Scavenging Activity

The radical scavenging activity of wheatgrass powder was determined according to Kashyap et al. [15] using 2,2-diphenyl-1-picrylhydrazyl (DPPH). The antioxidant activity was measured by
$$\mathrm{Antioxdant}\;\mathrm{activity}\left(\%\right)=\frac{A-B}{A}\times 100$$
where A is the absorbance of the DPPH solution and B is the absorbance of the sample in the DPPH solution.

#### 2.6.5. Ferric Reducing Antioxidant Power (FRAP)

The FRAP assay of wheatgrass powder was performed according to Kashyap et al. [15]. First, 0.1mL of the sample was diluted using distilled water and added to 3mL of the FRAP reagent (300 mM acetate buffer (pH 3.6), 0.03 mg of TPTZ in a 10:1:1 ratio with 40 mMHCl and 20 mM ferric chloride) and allowed to stand for 30 min in the dark. The absorbance was measured at 593 nm. Ascorbic acid was taken as standard, and the data were expressed in mgAAE/g.

#### 2.6.6. Rehydration Ratio

The rehydration ratio (RR) was performed according to Mc Minn and Magee [16]. A 5 g sample of powdered wheatgrass was added to 150 mL of distilled water in a beaker. This mixture was boiled for approximately three minutes by covering it with a watch glass. After 5 min of rehydration, the water was removed by transferring the mixture into a Buchner funnel previously covered with Whatman filter paper No. 4. A slight vacuum was applied to remove excess water. The samples were then taken out and weighed. The rehydration ratios were calculated using the equation below:$$\mathrm{RR}=\frac{\mathrm{Mrh}}{\mathrm{Mdh}}$$
where Mrh denotes the weight of the sample which was rehydrated (kg) and Mdh denotes the weight of the sample after drying (kg).

#### 2.6.7. Water Activity

The water activity was measured using a water activity meter (Rotronic water activity meter, Cole-Parmer, Mumbai, India).

#### 2.6.8. Functional Properties

##### Water Binding Capacity (WBC)

WBC was assessed according to Elaveniya and Jayamuthunagai [17]. A powdered sample of 1 g was dissolved in 10 mL of distilled water. After stirring for 1hour, the mixture was centrifuged at 2200 rpm for 15 min. WBC was calculated based on the weight of the residue after centrifugation.

#### 2.6.9. Oil Binding Capacity (OBC) 

Similar to WBC, OBC was determined, but canola oil was replaced with distilled water, and the results were calculated as 1 g of oil per 100 g of sample [17].

##### Solubility and Swelling Power

Solubility and Swelling Power were measured according to the method of Elaveniya and Jayamuthunagai [17]. The sample (1 g) was poured into a pre-weighed centrifuge tube, and 10 mL of distilled water was mixed. The centrifuge tube was kept in a water bath shaker at 80 °C for 30 min. It was then removed from the water bath and placed at room temperature for cooling, after which it was centrifuged at 2200 rpm for 15 min. The residue was collected, dried, and weighed by evaporating the supernatant to find out the solubility of the sample. The solubility was calculated using the following formula:$$\mathrm{Solubility}\;\left(\%\right)=\frac{\mathrm{Weight}\;\mathrm{of}\;\mathrm{dried}\;\mathrm{sample}\;\mathrm{in}\;\mathrm{supernatant}}{\mathrm{Weight}\;\mathrm{of}\;\mathrm{the}\;\mathrm{original}\;\mathrm{sample}}\times 100$$

In order to determine the swelling power, the supernatant was decanted after centrifugation and the sample paste weighed. The swelling power was calculated using the formula:$$\mathrm{Swelling}\;\mathrm{Power}=\frac{\mathrm{Weight}\;\mathrm{of}\;\mathrm{wet}\;\mathrm{mass}\;\mathrm{sediment}}{\mathrm{Wet}\;\mathrm{of}\;\mathrm{dry}\;\mathrm{matter}\;\mathrm{in}\;\mathrm{the}\;\mathrm{gel}}$$

##### Foam Capacity and Foam Stability

The foam capacity and foam stability were measured according to Elaveniya and Jayamuthunagai [17]. A 2 g sample was mixed with 50 mL of distilled water in a 100 mL measuring cylinder. The suspension was whipped vigorously for foam formation, and the volume was noted. The foam capacity was calculated as:$$\mathrm{Foam}\;\mathrm{Capacity}\;\%=\frac{\mathrm{Volume}\;\mathrm{after}\;\mathrm{whipping}-\mathrm{Volume}\;\mathrm{before}\;\mathrm{whipping}}{\mathrm{Volume}\;\mathrm{before}\;\mathrm{whipping}}\times 100$$

To determine the foam stability, the foam volume was noted after 1 h of whipping as a percentage of the initial foam volume:$$\mathrm{Foam}\;\mathrm{Stability}\;\%=\frac{\mathrm{Foam}\;\mathrm{volume}\;\mathrm{after}\;\mathrm{standing}\;\mathrm{time}\;\left(60\;\mathrm{minutes}\right)}{\mathrm{Initial}\;\mathrm{foam}\;\mathrm{volume}}\times 100$$

### 2.7. Statistical Analysis

All tests were conducted in triplicate, and the results are presented as mean ± standard deviation (SD). The data were analysed with an analysis of variance (ANOVA) by Duncan’s multiple range test (*p* ≤ 0.05) using Statistica-10 (M/s. Stat Soft Inc., St Tulsa, OK, USA).

## 3. Results and Discussion

### 3.1. Analysis of Drying Curve

The temperature-dependent changes in the moisture content of wheatgrass during its time exposed to air are shown in Figure 1. The obtained characteristic drying curves for the drying of wheatgrass at 50 to 70 °C with an interval of 5 °C were found similar to those of most agricultural products and confirmed the temperature dependence of moisture loss [12]. The moisture content was found to be reduced exponentially during the operation as drying time increased for wheatgrass, but the samples’ moisture was reduced at a faster rate as the temperature reached 70 °C. The drying time for wheatgrass samples was 180, 150, 120, 90, and 60 min at 50, 55, 60, 65, and 70 °C, respectively, at a fixed bed height. The moisture evaporation rate increases and therefore the moisture ratio decreases with all drying temperatures and times. From Figure 1, it is also clear that the drying time decreases with increased temperature. This indicates that wheat grass’s drying kinetics are affected by the temperature of the air while drying. 

Similar results were observed during the dehydration of various fruits and vegetables, such as drying eggplant in a fluidized bed drier for 30, 25, and 20 min at 60, 70, and 80 °C [18], and drying olive pomace for 135, 100, 60 and 45 min at 50, 60, 70, and 80 °C [12].

### 3.2. Analysis of Drying Rate

Figure 2 illustrates the relationship between temperature and drying rate for wheatgrass at different air-drying temperatures as a function of the moisture content.

Initially, there was a high drying rate, but it declined with the decreasing moisture content and drying time. The drying experiment did not demonstrate any constant rate of drying. At all the tested dehydration temperatures, wheatgrass was drying at a falling rate, further indicating the importance of diffusion in the transfer of moisture or mass. Similar results were observed in the case of treated and untreated plums [19] and apple pomace [20]. The low moisture content prevents water from moving to the surface and reduces the amount of water that evaporates from the surface over time. The initial drying rates were highest at 70 °C and lowest at 50 °C. This indicates that moisture was removed faster from the sample at high inlet air temperatures for the same initial moisture content [18].

### 3.3. Modeling of Drying Curve

The moisture content data while drying were used to calculate a dimensional moisture ratio that could be fitted to selected basic dehydration models and represent the dehydration behaviour of biomaterials based on the calculated dimensional moisture ratio. The initial moisture content of the model was assumed to be a critical moisture content for fitting purposes. The moisture content data obtained from the drying experiment were fitted to eight mathematical models (Newton, Page, Henderson and Pabis, Modified Henderson and Pabis, Logarithmic, two-term, Diffusion approach Midili et al. [21]), as listed in Table 1. 

**Table 1 foods-12-01576-t001:** Drying models applied to wheatgrass samples’ drying curves. Source: Hawa et al. [22].

S. No.	Model Name	Model
1.	Newton	MR = exp (−kt)
2.	Page Model	MR = exp (−kt^n^)
3.	Modified Page Model	MR = exp (−kt)
4.	Henderson and Pabis	MR = a exp (−kt)
5.	Modified Henderson and Pabis	MR = a exp (−kt) + b exp (−gt) + c
6.	Logrithmic	MR = a exp(−kt) + C
7.	Singh Model	MR = exp(−kt) − akt
8.	Midiliet al. [19]	MR = a exp (−kt^n^) +bt
9.	Weibull Model	MR = exp(−(t/a)^b^)
10.	Two-term model	MR = a exp (−k_o_t) + b exp (−k_1_t)
11.	Two-term exponential model	MR = a exp(−kt) + (1 − a) exp(kat)
12.	Wang and Singh Model	MR = 1 + at + bt^2^

Fitted parameters were validated by selecting those with the highest R^2^ values and the lowest χ^2^ and RMSE. The R^2^, RMSE, and χ^2^ values varied from 0.994 to 0.999, 0.0134 to 0.5983, and 0.000001 to 0.00136, respectively (Table 2). Out of all the models, the Page and Logarithmic models gave the highest R^2^ values and lowest χ^2^ and RMSE. Wheatgrass drying appears to be best described by these two models. It was also found that wheatgrass dried at 60 °C had the highest R^2^ value and the lowest χ^2^ and RMSE values. Doymaz et al. [23] and Meziane [12] used the Page model and the Handerson and Pabis model to stimulate the drying of olive pomace, and it was found that the Page model provided the best fit to moisture content. Very low χ^2^ and RMSE values further confirm the applicability of the Page model in representing the effective reproducibility of the drying process.

### 3.4. Effective Moisture Diffusivity and Activation Energy

It was found that wheatgrass drying occurred during falling rate period. Throughout the whole drying process, the effective moisture diffusivity (D_eff_) was determined as an arithmetic average at decreasing moisture contents. The D_eff_ increased with increasing drying temperature and in the range of 1.23–2.81 × 10^−10^m^2^/s. The values of D_eff_ are similar to the earlier research, where D_eff_ values for drying of olive pomace were 0.68–2.15 × 10^−7^ m^2^/s in the temperature range of 50–60 °C [12], 4.95 × 10 ^−10^–1.42 × 10^−7^ m^2^/s in the temperature range of 50–110 °C [24] for strawberry, and 5.9902 × 10^−8^–2.6616 × 10^−7^ m^2^/s in the temperature range of 60–120 °C [25] for coconut for all in Fluidized Bed Driers.

Arrhenius-type equations can be used to describe the temperature dependence of effective moisture diffusivity. A plot of the slope of ln (D_eff_) against the reciprocal of temperature, 1/(T + 273.15), was used to calculate the activation energy (Ea). The value of the activation energy of wheatgrass is 34.535 KJ/mol. Similar values of activation energy have been mentioned in previous studies, where they found the drying of potato slices ranged from 39.49 to 42.34 kJ/mol for different bed heights [26]. In another study of olive pomace drying, the activation energy values ranged from 34.05 to 36.84 and 38.10 kJ/mol for 41 mm, 52 mm, and 63 mm bed heights, respectively [12].

### 3.5. Effect of Different Air-Drying Temperatures on Proximate Composition of Wheatgrass Powder

Table 3 shows the effect of different drying temperatures of air on the proximate composition of wheatgrass powder. It was seen that drying temperature does not change the protein, fat, ash, carbohydrate, and fibre content, but the moisture of the wheatgrass powder showed significant changes. Wheatgrass has an initial moisture content of 90.44 ± 0.14%. When the temperature was raised from 50 °C to 70 °C, the moisture content of powdered samples significantly decreased. This could be due to a rise in thermal energy because of higher temperatures, which also results in the vaporization of moisture at higher air-drying temperatures, thus lowering the relative humidity of the air allowing more moisture to be removed from the sample [27]. A similar trend of decrease in moisture was supported by Ajala et al. [28]. Ogundeleolusola et al. [29] reported that the moisture content in (*Colocasiaesculenta*) Cocoyam flour was reduced with the rise in drying temperature. An insignificant change in carbohydrate, protein, fat, ash, and fibre content was seen in dried wheatgrass powder with increased temperature. As the drying temperature increases, more water is removed from the sample; therefore, the carbohydrate content will increase [30]. The same temperature effect on carbohydrates was reported earlier by Kakade and Hathan [31] in beetroot leaves and by Uribe et.al. [32] in olive waste cakes.

### 3.6. Effect of Different Air-Drying Temperatures on TPC, Antioxidant Activity, and Chlorophyll Content of Wheatgrass

Table 3 illustrates the TPC of wheatgrass powders dried at different temperatures. It has been observed that with the rise in drying temperature, the TPC of wheatgrass increased. However, the ANOVA results showed that a significant change in TPC occurred from 50 °C (117.16 ± 0.41 mgGAE/g) to 55 °C (127.63 ± 1.17 mgGAE/g), but no significant difference was seen with further increases in temperature. This might be due to the shorter drying time at higher temperatures because phenolic compounds are heat-sensitive and the long duration of heating can cause such irreversible chemical changes to phenolic compounds [33]. In addition, lower drying temperatures did not inactivate the antioxidant enzyme, i.e., polyphenol oxidase, and longer drying times caused a marked effect on the total phenolic content of wheatgrass. A similar finding was also noticed in the DPPH and FRAP assays while drying plums to powders [34] and in the case of drying blueberries [35]. The antioxidant activities are directly linked with the total phenolic contents [36] available in the wheatgrass extracts. Wheatgrass is loaded with polyphenols, which shows a relationship between antioxidant activity and ferric-reducing antioxidant power [37]. Antioxidant activity might also be attributed to the formation of novel compounds, such as products of the Maillard reaction, which possess antioxidant properties [38]. 

The chlorophyll content of wheatgrass powders dried at different air-drying conditions is given in Table 3. The chlorophyll content (6.11 ± 0.02 to 2.31 ± 0.03 mg/g) significantly decreased with elevated temperature. Brestic et al. [39] reported that if wheat leaves are exposed to higher temperatures, it can damage the structure and function of chloroplasts, which lowers the photosynthesis rate, and therefore the chlorophyll content is ultimately reduced. The results obtained were similar to the previous works reported by Alisher et al. [40], where the chlorophyll(a) content was seen to decrease in the three varieties of cotton leaves with an increase in temperature. 

### 3.7. Effect of Different Temperatures on Functional Properties, Rehydration Ratio, and Water Activity (a_w_) of Wheatgrass

The functional properties are the quality attributes of the food products, which determine the complexity of its physiochemical properties, structures, molecular conformation, and compositions during processing [41]. Since different studies have proposed that the preparation and cooking of foods can alter their functional properties, the drying process may also affect their functional properties [42]. Data on WBC, OBC, foam stability, foaming capacity, swelling power, and solubility of wheatgrass powder are presented in Table 4.

The WBCs of wheatgrass powdered samples dried at 50, 55, 60, 65, and 70 °C were 150.00 ± 2.00, 185.33 ± 6.80, 261.66 ± 2.51, 371.33 ± 2.51, and 222.66 ± 4.04%, respectively. It has been observed that wheatgrass WBC increases with the increase in air temperature. This might be due to the starch’s hydrophilic tendency, which had increased with an increase in air-drying temperature. Moreover, starch expands rapidly at higher temperatures, especially in the amorphous region. The WBC is an essential parameter for developing various food products, and a high absorption capacity may increase product cohesiveness. In addition, the functional properties of dried soybean powders showed a similar tendency to increase water-holding capacity with higher thermal and mechanical operations [43]. The polysaccharides and other compounds present in the powdered samples may undergo structural changes during the dehydration or drying process, and this is the most probable reason for showing a better WBC of the samples at higher drying temperatures [44]. Time and temperature are considered the most critical factors that ultimately describe polysaccharide stability [45].

The OBCs of wheatgrass powdered samples dried at 50, 55, 60, 65, and 70 °C were 523.33 ± 5.68, 564.00 ± 6.55, 596.00 ± 6.00, 612.66 ± 3.51, and 505.00 ± 7.00% respectively. The OBC of wheatgrass was seen to have increased with the rise in air temperature, but a significant difference was noticed only at 70 °C. This can be attributed to the information that the temperature rise could lower the moisture content and, hence, absorb the more of the oil. Oil absorption can improve the mouthfeel and retention of flavour by providing superior flavour and a soft texture to the food product [46]. The oil binding capacity was seen to increase with increased temperature in tamarind seed mucilage powder [47].

The foaming capacity of wheatgrass powder increases with a rise in drying temperature. This might be due to its high positive correlation with surface hydrophobicity. With the rise in temperature, the wheatgrass protein dispersions had greater surface hydrophobicity and lower surface tension, giving a better foaming capacity [48]. As a result of protein absorption at the interface, foam stability is greatly enhanced, and the foam is also more resistant to breaking down processes such as Ostwald ripening and coalescence ripening. The volume of foam decreases as the standing time increases. The high temperature might increase the protein aggregation and act as a pickeling-like stabilizer and food-grade emulsifier to stabilize the protein emulsion by forming a protein layer at the oil–water interface. Therefore, the foam stability can increase due to heat-induced protein aggregation [49]. Previous studies also had similar effects of temperature on foaming capacity and foaming stability in egg albumin [50]. 

The solubility of wheatgrass dried at temperatures of 50, 55, 60, 65, and 70 °C was 9.66 ± 0.57, 8.33 ± 0.57, 4.33 ± 0.57, 4.66 ± 0.57, and 3.33 ± 0.57%, respectively (Table 4). A rise in drying temperature might have caused structural modifications of substances that are affecting the polymers for the period of water removal, decreasing solubility. The difficulty in solubility occurs when high-solid foods are exposed to high temperatures. The solubility of most fruit and vegetable powders are meant for rehydration, and a powder which would set thoroughly and quickly without forming lumps would be ideal. Additionally, a previous study indicated that banana powder’s solubility decreased with increasing temperature [51].

The swelling powers of wheatgrass powdered samples dried at 50, 55, 60, 65, and 70 °C were 1.31 ± 0.006, 1.45 ± 0.004, 3.15 ± 0.007, 4.06 ± 0.006, and 3.838 ± 0.03%, respectively. Swelling power increased with temperature, and its effect can be directly related to the water binding capacity. The swelling power shows the strength and character of the starch granules present in the sample. The increase in the air-drying temperature has reduced the rehydration ratio of wheatgrass powder, as presented in Table 4. The rehydration ratios of wheatgrass leaves powder dried at 50, 55, 60, 65, and 70 °C were 11.64 ± 0.08, 10.09 ± 0.02, 9.13 ± 0.003, 8.50 ± 0.02, and 8.28 ± 0.03, respectively. The rehydration ratio is the property that indicates the powder’s ability to rehydrate and measures the damage to tissues that may have been caused during drying. The structural modifications related to food damage, usually at high temperatures, are loss of dense structure and integrity of tissues and a decrease in hydrophilic properties associated with the shrinkage of capillaries [52]. Similar trends of the rehydration ratio in the case of beetroot leaf powder were seen in previous works at air-drying temperatures of 50, 60, 70, and 80 °C, and the rehydration ratios were 3.36, 3.09, 2.01, and 0.119, respectively [31].

The water activity of wheatgrass powders that were dried at different air temperatures is represented in Table 4. The range of water activity of the powders was 0.27 ± 0.001–0.16 ± 0.000. The water activity was seen to have decreased with increasing temperature. The pattern of the data was similar to the previous research, where the water activity of dried papaya leaves was found to be highest at 50 °C (unblanched), which was 0.516, and the lowest water activity of 0.42 was found in a powder dried at 120 °C (unblanched) [53].

## 4. Conclusions

The drying kinetics of wheatgrass in a fluidized bed dryer were studied at temperatures between 50 and 70 °C at intervals of 5 °C. A linear decrease in the moisture content of wheatgrass with time was observed at the early stages of drying, but the trend became non-linear when the moisture content reached about 60–70% dry basis. At lower drying temperatures, there was little change in drying rates, and constant rate drying was not observed. The increased drying temperature also sped up the drying process, resulting in a shorter drying time. The range of D_eff_ was dependent on temperature, which was 1.23–2.81 × 10^−10^ m^2^/s, and the activation energy was 34.535 KJ/mol. The fitting parameter results for the basic drying model confirm the suitability of the Page and Logarithmic models in predicting the drying kinetics among the other selected models. However, the Page model appeared to be the most effective at predicting the drying kinetics at the five different temperatures based on the highest R^2^ and the reduced χ^2^ and RMSE. Based on the results of the proximate analysis, it appears that wheatgrass contains high levels of macronutrients, with no significant loss making them suitable for use in food products. Moreover, the powders’ functional properties, TPC, and antioxidant properties increased with the elevation of drying air temperature, while the chlorophyll content showed a steep decrease with the rise in temperature. In addition, a reduction in the total phenolic content and antioxidant activities were noticed at higher drying temperatures (65 and 70 °C). This showed that the stability of physicochemical parameters such as functional properties, TPC, antioxidant activities, and chlorophyll content of the dehydrated wheatgrass powders obtained after drying was primarily dependent on the temperature used during the drying experiment. The overall results obtained during the study of the drying kinetics of wheatgrass suggested that the drying temperature for the dehydration of wheatgrass should be 60 °C to retain all the valuable nutrients and stability of therapeutic compounds present in the product. Moreover, wheatgrass powder could be considered an important functional food that could be used to enhance therapeutic values and make a food product rich invaluable nutrients, as the powder’s quality was improved for a wider range of acceptability. Recommendations for future work include the consideration of cost and energy management in addition to drying time and product quality for the development of advanced fluidized bed drying technology, as these factors will help in the transition from laboratory to industrial scales. 

## Figures and Tables

**Figure 1 foods-12-01576-f001:**
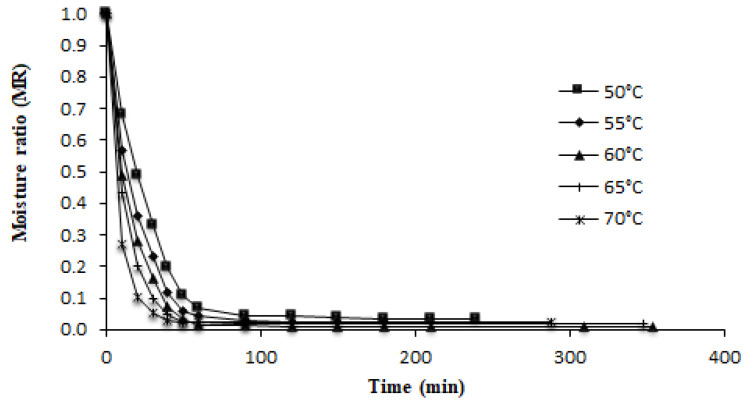
Moisture ratio vs. drying time at different air temperatures for wheatgrass.

**Figure 2 foods-12-01576-f002:**
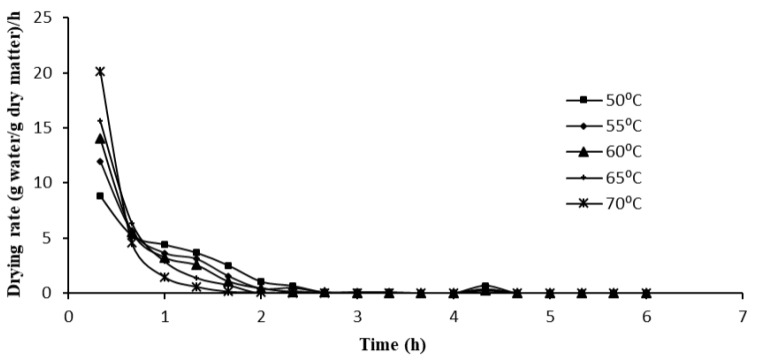
Changes in drying rate with drying time at different air temperatures for wheatgrass.

**Table 2 foods-12-01576-t002:** Statistical results of eight mathematical models at different drying conditions and constant bed height.

Model	Temperature (°C)	R^2^	χ^2^	RMSE
Newton	50	0.998185	0.000188	0.013215
	55	0.997649	0.000348	0.01777
	60	0.998185	0.000188	0.040223
	65	0.99502	0.001363	0.042966
	70	0.998456	0.000519	0.059836
Page	50	0.995465	0.000786	0.013484
	55	0.99748	0.000336	0.014148
	60	0.998573	0.000136	0.013215
	65	0.999292	0.00014	0.014148
	70	0.998862	0.0002	0.015058
Henderson and Pabis	50	0.993888	0.000852	0.018654
	55	0.997481	0.000378	0.017777
	60	0.998114	0.000235	0.017777
	65	0.999332	0.000197	0.017777
	70	0.998386	0.000591	0.016476
Logarithmic	50	0.994888	0.000707	0.017777
	55	0.998096	0.00023	0.017777
	60	0.998246	0.000225	0.017777
	65	0.999844	0.0000179	0.017777
	70	0.999585	0.000057	0.016476
Two Term	50	0.99454	0.000963	0.017777
	55	0.997071	0.000434	0.017777
	60	0.998138	0.000266	0.017777
	65	0.999304	0.000189	0.017777
	70	0.999631	0.0000607	0.016476
Diffusion Approach	50	0.995642	0.000629	0.017777
	55	0.998239	0.000216	0.017777
	60	0.99824	0.000232	0.017777
	65	0.999847	0.0000176	0.017777
	70	0.999631	0.0000508	0.016476
Modified Henderson and Pabis	50	0.995871	0.00077	0.017777
	55	0.998237	0.000284	0.017777
	60	0.998958	0.000173	0.017777
	65	0.999883	0.0000179	0.017777
	70	0.999993	0.00000135	0.016476
Midilli et al.	50	0.99538	0.000907	0.017777
	55	0.997448	0.000434	0.017777
	60	0.998526	0.000205	0.017777
	65	0.999279	0.000178	0.017777
	70	0.999197	0.000233	0.016476

**Table 3 foods-12-01576-t003:** Effect of drying temperature on proximate composition, chlorophyll content, total phenolic content, and antioxidant activity of wheatgrass.

Drying Temp.	Moisture Content (%)	Protein (%)	Fat (%)	Ash (%)	Crude Fibre (%)	Carbohydrate (%)	Chlorophyll Content (mg/g)	Total Phenol Content (mgGAE/g)	Antioxidant Activity	
									DPPH (%)	FRAP (mgAAE/g)
50 °C	5.26 ± 0.04 ^a^	24.54 ± 0.04 ^a^	2.41 ± 0.05 ^a^	7.86 ± 0.02 ^a^	12.22 ± 0.04 ^a^	47.71 ± 0.01 ^a^	6.11 ± 0.02 ^a^	117.16 ± 0.41 ^b^	33.56 ± 0.08 ^b^	1.474 ± 0.004 ^b^
55 °C	5.28 ± 0.05 ^b^	24.63 ± 0.07 ^a^	2.40 ± 0.02 ^a^	7.85 ± 0.02 ^a^	12.23 ± 0.03 ^a^	47.68 ± 0.05 ^a^	6.14 ± 0.05 ^a^	127.63 ± 1.17 ^a^	37.19 ± 0.10 ^a^	1.587 ± 0.001 ^a^
60 °C	4.81 ± 0.02 ^c^	24.67 ± 0.07 ^a^	2.39 ± 0.02 ^a^	7.87 ± 0.01 ^a^	12.20 ± 0.04 ^a^	47.65 ± 0.07 ^a^	5.49 ± 0.05 ^b^	128.10 ± 0.43 ^a^	37.21 ± 0.21 ^a^	1.617 ± 0.001 ^a^
65 °C	4.06 ± 0.02 ^d^	24.66 ± 0.01 ^a^	2.39 ± 0.02 ^a^	8.19 ± 0.02 ^a^	12.23 ± 0.05 ^a^	47.78 ± 0.08 ^a^	2.64 ± 0.03 ^c^	128.66 ± 1.16 ^a^	37.48 ± 0.08 ^a^	1.565 ± 0.009 ^b^
70 °C	4.01 ± 0.03 ^e^	24.91 ± 0.08 ^a^	2.40 ± 0.03 ^a^	7.87 ± 0.01 ^a^	12.19 ± 0.04 ^a^	47.61 ± 0.09 ^a^	2.31 ± 0.03 ^d^	128.53 ± 0.55 ^a^	34.80 ± 0.27 ^b^	1.372 ± 0.001 ^c^

Values are presented in mean ± SD with different superscript letters in a column being significantly different (*p* < 0.05).

**Table 4 foods-12-01576-t004:** Effect of drying temperature on functional properties, rehydration ratio, and water activity of wheatgrass.

Drying Temp.	WBC (%)	OBC (%)	FC (%)	FS (%)	Swelling Power (%)	Solubility (%)	RR	a_w_
50 °C	150.00 ± 2.00 ^e^	523.33 ± 5.68 ^d^	37.99 ± 1.14 ^b^	46.53 ± 1.10 ^d^	1.31 ± 0.006 ^e^	3.33 ± 0.57 ^c^	11.64 ± 0.08 ^a^	0.27 ± 0.001 ^a^
55 °C	185.33 ± 6.80 ^d^	564.00 ± 6.55 ^c^	43.47 ± 0.71 ^a^	70.52 ± 1.44 ^b^	1.45 ± 0.004 ^d^	4.33 ± 0.57 ^c^	10.09 ± 0.02 ^b^	0.21 ± 0.000 ^b^
60 °C	261.66 ± 2.51 ^b^	596.00 ± 6.00 ^b^	42.41 ± 0.70 ^a^	59.75 ± 1.42 ^c^	3.15 ± 0.007 ^c^	8.33 ± 0.57 ^b^	9.13 ± 0.003 ^c^	0.19 ± 0.001 ^b,c^
65 °C	371.33 ± 2.51 ^a^	612.66 ± 3.51 ^a^	43.77 ± 0.40 ^a^	77.22 ± 3.17 ^a^	4.06 ± 0.006 ^a^	9.66 ± 0.57 ^a^	8.50 ± 0.02 ^d^	0.16 ± 0.001 ^c^
70 °C	222.66 ± 4.04 ^c^	505.00 ± 7.00 ^e^	21.79 ± 1.24 ^c^	76.63 ± 1.71 ^a^	3.83 ± 0.03 ^b^	10.33 ± 0.57 ^a^	8.28 ± 0.03 ^e^	0.16 ± 0.000 ^c^

Values are presented in mean ± SD with different superscript letters in a column being significantly different (*p* < 0.05) WBC: Water binding capacity; OBC: Oil binding capacity; FC: Foaming capacity; FS: Foaming stability; RR: Rehydration ratio; a_w_: Water activity.

## Data Availability

Not applicable.

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
