# Peer review of "Fluidized Bed Drying of Wheatgrass: Effect of Temperature on Drying Kinetics, Proximate Composition, Functional Properties, and Antioxidant Activity"

_foods, 2023, doi:10.3390/foods12081576_

Round 1

Reviewer 1 Report

Overall, the manuscript is not clearly presented. A large number of indicators add up, which makes the topic of the article is not prominent. For example, chlorophyll content, ash, fat and so on seem irrelevant to the topic of the article. Furthermore, some functional indicators...

The Introduction is not clear and logical. It does not reflect the necessity and urgent need to study the effect of Fluidized bed dry on Wheatgrass.

Line20-34, line490-518, Abstract and Conclusion do not summarize the  manuscript well.

 Line174: Detail sampling information is not described in this section. In Table 3, there were significant differences in water content among the groups (more than 1.2%). It seems that sampling is not the end of drying, which may also cause differences in other indicators.

Some wrong format, such as line277-280: Figure 1 is separated from the notes; . Same in line289-291.

Line386-389:Lack of reference.

There are a number of points which must be improved. To sum up, the paper does not meet average scientific standards and publication cannot be recommended by the reviewer. 

Author Response

Reviewer 1

Overall, the manuscript is not clearly presented. A large number of indicators add up, which makes the topic of the article is not prominent. For example, chlorophyll content, ash, fat and so on seem irrelevant to the topic of the article. Furthermore, some functional indicators...

Reply:

Comment: The Introduction is not clear and logical. It does not reflect the necessity and urgent need to study the effect of Fluidized bed dry on Wheatgrass.

 Reply: Thank you for your suggestions and as suggested necessary changes has been incorporated in the introduction section to better understand the need of using fluidized bed drying on wheatgrass.

Comment: Line20-34, line490-518, Abstract and Conclusion do not summarize the manuscript well.

 Reply: Thank you for your valuable suggestions and asper the suggestion changes has been done in the abstract and conclusion part of revised manuscript.

Comment: Line174: Detail sampling information is not described in this section. In Table 3, there were significant differences in water content among the groups (more than 1.2%). It seems that sampling is not the end of drying, which may also cause differences in other indicators.

 Reply: Detail sampling has been incorporated in the revised manuscript.

This change in the moisture content with increasing temperature is because the higher temperatures results in the vaporization of moisture and, the relative humidity of the air will be lower and so more will be the removal of moisture from the sample. As the difference in the moisture content was not that much high so no significant difference was seen in the proximate composition of wheatgrass but other biochemical properties were significantly affected by the temperature.

Comment: Some wrong format, such as line277-280: Figure 1 is separated from the notes; . Same in line289-291.

 Reply: Changes has been done in the revised manuscript.

Comment: Line386-389:Lack of reference.

 Reply: The reference has been added for the given sentence in the revised manuscript.

Reviewer 2 Report

Authors evaluated the effect of fluidized bed drying on proximate, antioxidant and functional characteristics of wheatgrass. In general, the data and results have scientific merits, but the major concern is lack of in-depth discussion. If possible, please provide a brief explanation/ hypothesis of the mechanism/relationship behind the facts, rather than simply describing these phenomena. Furthermore, there are some specific comments.

(1) In the Introduction, the authors should highlight as to why the fluidized bed drying was employed for the study. Compared with other drying methods, the rationale for fluidized bed drying needs to be emphasized. Moreover, the published papers about the fluidized bed drying are suggested to supplement in this section. Also, authors need to focus on new findings in this study.

(2) Please specify the determination method of proximate composition.

(3) Many sentences are ambiguous and dyslexic. For example, Lines 210-212, ‘this mixture was then added to 3ml of FRAP reagent (300mM acetate buffer (pH 3.6), TPTZ (0.03mg in 10ml 40mM HCl) and 20mM ferric chloride in 10:1:1 ratio) and the mixture and was then allowed to stand in the dark for 30 minutes.’, Line359, ‘The moisture content which was initially present in fresh wheatgrass was 90.44±0.14 %’, and so on. The authors need to revise them.

(4) Line 365, ‘Ogundeleo-365 lusola et al. reported that the moisture content in (Colocasia esculenta) Cocoyam flour was reduced with the rise in drying temperature’, which is different from the results in this paper. Please explain this phenomenon.

(5) In Section 3.6, ‘the shorter drying time at higher temperatures because phenolic compounds are heat sensitive and the long duration of heating can cause such chemical changes to irreversible phenolic compounds’, please supplement the references.

(6) What are the future research work which can be done based on these study outcomes? The application of the study outcomes needs to be explained in detail towards the concluding paragraphs of the write-up in the end of the manuscript.

Author Response

Reviewer 2

Comment: In the Introduction, the authors should highlight as to why the fluidized bed drying was employed for the study. Compared with other drying methods, the rationale for fluidized bed drying needs to be emphasized. Moreover, the published papers about the fluidized bed drying are suggested to supplement in this section. Also, authors need to focus on new findings in this study.

Reply: Thank you for the suggestions and as suggested the comparison of fluidized bed drying with other method methods had been incorporated. Furthermore, study related to fluidized bed drying has also been added in the introduction section.

Comment: Please specify the determination method of proximate composition.

Reply: The determination methods of proximate composition has been incorporated in the revised manuscript.

Comment: Many sentences are ambiguous and dyslexic. For example, Lines 210-212, ‘this mixture was then added to 3ml of FRAP reagent (300mM acetate buffer (pH 3.6), TPTZ (0.03mg in 10ml 40mM HCl) and 20mM ferric chloride in 10:1:1 ratio) and the mixture and was then allowed to stand in the dark for 30 minutes.’, Line359, ‘The moisture content which was initially present in fresh wheatgrass was 90.44±0.14 %’, and so on. The authors need to revise them.

Reply: As suggested by the reviewer, the lines has been revised.

Comment: Line 365, ‘Ogundeleo-365 lusola et al. reported that the moisture content in (Colocasia esculenta) Cocoyam flour was reduced with the rise in drying temperature’, which is different from the results in this paper. Please explain this phenomenon.

Reply: The results of this study also showing that moisture content decreased with increasing drying temperature as shown in Table 3. The reason for this could be due to a rise in thermal energy because of higher temperatures, which results in the vaporization of moisture also at higher air-drying temperatures, the relative humidity of the air will be lower and so more will be the removal of moisture from the sample.

The same has already been incorporated in the manuscript.

Comment: In Section 3.6, ‘the shorter drying time at higher temperatures because phenolic compounds are heat sensitive and the long duration of heating can cause such chemical changes to irreversible phenolic compounds’, please supplement the references.

Reply: The reference has been incorporated for the given line in the revised manuscript.

Comment: What are the future research work which can be done based on these study outcomes? The application of the study outcomes needs to be explained in detail towards the concluding paragraphs of the write-up in the end of the manuscript.

Reply: Thank you for the suggestions and as suggested the future work and applications sections has been incorporated in the last part of the conclusion section.

Reviewer 3 Report

This study investigated the effect of fluidized bed drying on proximate, antioxidant and functional characteristics of wheatgrass. There are good application and science value. However, there are some significant issues that should be carefully analysed.

1) The "KJ/mol" is not written correctly in Line 27. It should be "kJ/mol".

2) There is no formula of "the coefficient of determination" in the text in Line 136, which should be supplemented.

3) In line 153, "effective diversity" is represented as "De", but in line 338, it is represented as "Deff". The front and back expressions are inconsistent and need to be modified.

4) Figure 1 and title are not in the same place in Line 280.

5) The data in Figures 1 and 2 are the average of multiple measurements or the word measurement results. It needs to be explained in the text.

6) The writing format of "R2" is incorrect in Line 320.

7) The font format of " Triticum Aestivum L" should be in italics in Line 536.

8) The font format of " Triticum Aestivum " should be in italics in Line 544.

9) The font format of " Cymbopogon Citratus Stapf " should be in italics in Line 549.

10) The font format of " Solanum Melongena" should be in italics in Line 569.

11) The font format of " Piper Retrofractum Vahl" should be in italics in Line 578.

12) The font format of " Lagenaria Siceraria" should be in italics in Line 588.

13) The font format of " Theobroma Cacao" should be in italics in Line 590.

14) The font format of " Vaccinium Corymbosum L. " should be in italics in Line 605.

15) The font format of " Prunus Nepalensis " should be in italics in Line 609.

16) The font format of " Aloe Barbadensis " should be in italics in Line 626.

17) The font format of " Tamarindus Indica L. " should be in italics in Line 634.

18) The font format of " Saccharina Latissima " should be in italics in Line 642.

19) The font format of " Colocasiaesculenta " should be in italics in Line 593. There are similar writing format problems. I hope the author will carefully check and revise them.

Author Response

Reviewer 3

Comment: The "KJ/mol" is not written correctly in Line 27. It should be "kJ/mol".

Reply: Changes has been incorporated in the revised manuscript.

Comment: There is no formula of "the coefficient of determination" in the text in Line 136, which should be supplemented.

Reply: Thank you for your suggestion as as suggestion same has been incorporated in the revised manuscript.

Comment: In line 153, "effective diversity" is represented as "De", but in line 338, it is represented as "Deff". The front and back expressions are inconsistent and need to be modified.

Reply: As suggested the required changes has been incorporated in the revised manuscript.

Comment: Figure 1 and title are not in the same place in Line 280.

Reply: Changes has been incorporated in the revised manuscript.

Comment: The data in Figures 1 and 2 are the average of multiple measurements or the word measurement results. It needs to be explained in the text.

Reply: The data presented in the both the figures are the mean data of multiple measurements and in the section 2.7, it has been mentioned that all the experiments were performed in the triplicates and represented as mean.

Comment: The writing format of "R2" is incorrect in Line 320.

Reply: Changes has been incorporated in the revised manuscript.

Comment: The font format of " Triticum Aestivum L" should be in italics in Line 536.

Reply: Changes has been incorporated in the revised manuscript.

Comment: The font format of " Triticum Aestivum " should be in italics in Line 544.

Reply: Changes has been incorporated in the revised manuscript.

Comment: The font format of " Cymbopogon Citratus Stapf " should be in italics in Line 549.

Reply: Changes has been incorporated in the revised manuscript.

Comment: The font format of " Solanum Melongena" should be in italics in Line 569.

Reply: Changes has been incorporated in the revised manuscript.

Comment: The font format of " Piper Retrofractum Vahl" should be in italics in Line 578.

Reply: Changes has been incorporated in the revised manuscript.

Comment: The font format of " Lagenaria Siceraria" should be in italics in Line 588.

Reply: Changes has been incorporated in the revised manuscript.

Comment: The font format of " Theobroma Cacao" should be in italics in Line 590.

Reply: Changes has been incorporated in the revised manuscript.

Comment: The font format of " Vaccinium Corymbosum L. " should be in italics in Line 605.

Reply: Changes has been incorporated in the revised manuscript.

Comment: The font format of " Prunus Nepalensis " should be in italics in Line 609.

Reply: Changes has been incorporated in the revised manuscript.

Comment: The font format of " Aloe Barbadensis " should be in italics in Line 626.

Reply: Changes has been incorporated in the revised manuscript.

Comment: The font format of " Tamarindus Indica L. " should be in italics in Line 634.

Reply: Changes has been incorporated in the revised manuscript.

Comment: The font format of " Saccharina Latissima " should be in italics in Line 642.

Reply: Changes has been incorporated in the revised manuscript.

Comment: The font format of " Colocasiaesculenta " should be in italics in Line 593. There are similar writing format problems. I hope the author will carefully check and revise them.

Reply: Changes has been incorporated in the revised manuscript.

Sozzi, A.; Zambon, M.; Mazza, G.; Salvatori, D. Fluidized bed drying of blackberry wastes: Drying kinetics, particle characterization and nutritional value of the obtained granular solids. Powder technol. 2021, 385, 37-49.

Subramaniam, S.D.; Mudalip, S.K.A.; Halim, L.A.; Basrawi, F.; Azman, N.A.M. Evaluation of the Drying Kinetics of Swirling Fluidized Bed Drying and Performance Assessment on the Nutritional Properties of Bentong Ginger. 2022, preprints, 1-10.

Sarjerao, L.S.K.; Kashyap, P.; Sharma, P. Effect of drying techniques on drying kinetics, antioxidant capacity, structural, and thermal characteristics of germinated mung beans (Vigna radiata). J.Food Process Eng. 2022, 45(11), e14155.
